# Remyelination in Multiple Sclerosis: Findings in the Cuprizone Model

**DOI:** 10.3390/ijms232416093

**Published:** 2022-12-17

**Authors:** Heinig Leo, Markus Kipp

**Affiliations:** Institute of Anatomy, Rostock University Medical Center, 18057 Rostock, Germany

**Keywords:** cuprizone, multiple sclerosis, remyelination, demyelination, progression, protection

## Abstract

Remyelination therapies, which are currently under development, have a great potential to delay, prevent or even reverse disability in multiple sclerosis patients. Several models are available to study the effectiveness of novel compounds in vivo, among which is the cuprizone model. This model is characterized by toxin-induced demyelination, followed by endogenous remyelination after cessation of the intoxication. Due to its high reproducibility and ease of use, this model enjoys high popularity among various research and industrial groups. In this review article, we will summarize recent findings using this model and discuss the potential of some of the identified compounds to promote remyelination in multiple sclerosis patients.

## 1. Introduction

The nervous system is simultaneously complicated and fascinating. No other scientific field has recorded more advances recorded in the last few decades than neuroscience. We owe these great advances to the new technologies that have been recently developed, enabling us to visualize and manipulate central nervous system (CNS) cells, as well as the development of appropriate in vivo [1] and in vitro [2] models to study CNS health and disease at the molecular level. Using these models, we can now study both tissue destruction and tissue repair.

The cells of the nervous system can be divided into nerve cells (neurons) and glial cells. Neurons are responsible for signal transmission by generating action potentials and passing them on to communicating neurons. The place where nerve cells communicate with each other is called the synapse. It is textbook knowledge that the central and peripheral nervous system consist not only of neurons, but also of other cells that are morphologically and functionally different from neurons. The co-discoverer of these non-neuronal cells in the mid-19th century, Rudolf Virchow, suspected a support and holding function, therefore naming them glial cells, derived from the Greek word glia for “glue”. Utilizing different staining methods by Santiago Ramón y Cajal, Pío del Río Hortega, and Camillo Golgi, glial cells were further subclassified at the end of the 19th century. The first differentiation of glial cells was made based on their size. Accordingly, microglia were distinguished from macroglia. The central macroglial cells include astrocytes, oligodendrocytes, and ependymal cells. This subclassification was wise: As is known today, microglial and macroglial cells have nothing to do with each other in terms of evolutionary history. Macroglial cells, i.e., oligodendrocytes, astrocytes, and ependymal cells, all originate from the neuroectoderm. In contrast, microglial cells represent immigrated blood cells; therefore, they originate from the mesoderm. Although the number of neurons in the human brain exceeds our imagination (about 100 billion), the number of glial cells exceeds that of neurons by a multiple. This review article will focus on the myelin-producing cells of the CNS, the oligodendrocytes, and their precursors.

The main function of oligodendrocytes and Schwann cells is the synthesis and maintenance of the myelin sheath, which is a lipid-rich biomembrane that spirals around the axons of most vertebrate nerve cells electrically insulating them. Compared to other biomembranes, myelin, which was discovered in 1854 by the pathologist Rudolf Virchow (1821–1902), has an exceptionally high lipid content (approximately 70% of dry weight) and a relatively low protein content (30%), with proteolipid protein (PLP) and myelin basic protein (MBP) being the most abundant ones [3]. Since myelin appears macroscopically white, highly myelinated regions in the CNS are also referred to as “white matter” in contrast to the less myelinated “gray matter” areas. Myelin sheaths along the axons are regularly interrupted by the nodes of Ranvier. Action potentials arise only at the nodes of Ranvier but not in the myelinated areas of the axon (id est, the internodes). This configuration allows for saltatory conduction, which is significantly faster than the continuous action potential propagation of non-myelinated fibers. In addition, this type of conduction saves energy, since an action potential only has to be built up at the location of the nodes and not continuously along an axon.

## 2. Remyelination Biology

There are various CNS disorders that are characterized by either dysmyelination or the destruction of a previously intact myelin sheath. Dysmyelination refers to a malformed and defective myelin sheath as opposed to the destruction of previously normal myelin that is seen in demyelinating conditions. In addition to the various forms of leukodystrophies, other genetically determined disorders, such as infantile amaurotic idiocy, hematosidosis, Niemann–Pick’s disease, and several of the aminoacidopathies are examples of dysmyelinating disorders [4]. The most frequent neurological disease with the central hallmark of myelin pathology is, however, multiple sclerosis (MS). In contrast to the above mentioned disorders, MS is a demyelinating condition in which intact myelin sheaths are destroyed by peripheral and central inflammatory cells. Pathological hallmarks, besides demyelination, are a focal breakdown of the blood–brain barrier (BBB), peripheral immune cell recruitment, neuronal and axonal damage, as well as microglia and astrocyte activation. During the initial phase of the disease, inflammation, mediated via the adaptive immune system, clinically results in specific behavioral deficits from which the affected patients can recover either entirely or partially. This disease phase is named ‘relapsing-remitting MS (RRMS)’. As the disease progresses, the frequency of new clinically detectable relapses decreases. Instead, there is a progressive accumulation of behavioral deficits from which the patients usually do not recover. This secondary disease phase is named ‘secondary progressive MS (SPMS)‘. Pathologically, SPMS is driven by a diffuse and chronic inflammatory process inside and around the brain and spinal cord parenchyma [5,6]. Eventually, patients initially present with a progressive disease course, which is named ‘primary progressive MS (PPMS)’.

Although MS is the most frequent demyelinating disease, white matter degeneration, oligodendrocyte dysfunction, or myelin destruction have been observed in other CNS disorders as well, such as Alzheimer’s disease [7], stroke [8,9], spinal cord injury [10], schizophrenia [11] or amyotrophic lateral sclerosis [12]. Consequently, cells of the oligodendrocyte lineage have been shown to exert important roles beyond those related to myelination, including regulation of angiogenesis in the normal postnatal brain [13], or antigen presentation and phagocytosis in mouse models of MS [14,15,16]. Remyelination is a multistep, complicated process that is very effective in young adults but loses effectiveness as one ages [17,18].

Since myelin debris is a potent inhibitor of remyelination, clearance of myelin debris by microglia and/or recruited monocytes in the first step is pivotal [19,20,21,22]. In a second step, due to the expanding (neuro-) inflammatory milieu, oligodendrocyte progenitor cells (OPCs) are activated, eventually proliferate [23], and migrate into the demyelinated area [24]. In the next series of molecular events, the OPCs differentiate into premyelinating oligodendrocytes, which change their morphology from a bipolar to a highly-branched, multipolar cell type. Finally, the premyelinating oligodendrocytes extend a process to the denuded axon that generates a new, lipid-rich, multi-lamellar myelin sheath. Oligodendrocyte proliferation and differentiation represents a delicate and complicated balance, and a number of factors have been identified that regulate these cellular events, including the Sonic Hedgehog signaling pathways [25], the Wnt (Wingless-type MMTV integration site family) signaling pathway [26], fibroblast growth factor 2 [27], different GPR family members [28], the transcription factor SOX2 [29], AKT (AKT serine/threonine kinase 1) [30] or BDNF (brain-derived neurotrophic factor) [31].

Several observations suggest that remyelination ameliorates functional deficits: Firstly, following experimental demyelination of the optic nerve, recovery in visual conductance is tightly correlated with MBP re-expression [32]. Secondly, accelerated remyelination via muscarinic cholinergic receptor knockdown prevents axonal loss in the experimental autoimmune encephalomyelitis (EAE) model [33]. Thirdly, the inhibition of remyelination, that is, experimentally induced by X-irradiation, results in a significant increase in the extent of axonal degeneration and loss compared to non-irradiated mice [34]. There appear to be multiple sources of remyelinating oligodendrocytes, including parenchymal OPC distributed ubiquitously throughout the gray and white matter, neural stem cells (NSCs) located in the subventricular zone(s) [35], and Schwann cells migrating from the periphery into the CNS [36].

## 3. Remyelination In Vivo Models

To study the complex physiology of remyelination and the factors involved in its failure, animal models are unavoidable. Although various species have been applied in the past in the context of myelin research, including dogs [37], cats [38], and zebrafishes [39], small rodents, especially mice, are the most frequently used. To study myelin degeneration and repair, toxin models are enjoying great popularity. In principle, toxin-mediated demyelination with subsequent remyelination can be induced using either focal injection of lysolecithin (also called lysophosphatidylcholine (LPC)) or ethidium bromide [40] into myelin-rich white matter tracts, or the systemic administration of the copper-chelator cuprizone. All three experimental approaches show robust endogenous remyelination after the demyelinating insult [41,42,43]. In this article, we will focus on the cuprizone model. First, we will describe the characteristics and histopathological changes of the model and then present an overview of factors that were identified to regulate myelin repair.

Cuprizone, chemically known as bis-(cyclohexanone) oxaldehydrozone, is a synthetic chelating compound initially used to detect trace copper [44,45]. This compound became of interest in biomedical research when it was discovered to exert toxic effects in the CNS of laboratory mice [46]. The intoxication of young adult mice with cuprizone, mixed into standard rodent chow in a concentration of 0.2–0.5% (*w*/*w*), induces, within some days, oligodendrocyte stress, leading to oligodendrocyte degeneration [47]. This presumably primary oligodendrocyte insult leads to the activation of astrocytes and microglia, the latter phagocytosing the degenerating myelin sheaths. This results in the acute demyelination of distinct white and gray matter brain regions [48]. Since cuprizone chelates copper and copper is an essential trace element for a number of metalloenzymes involved in cellular respiration, it is widely presumed that CNS damage due to the cuprizone intoxication is a result of copper dyshomeostasis and, in consequence, mitochondrial dysfunction. In contrast to this theory, results of a recent study suggest that cuprizone’s toxicity is not due to copper depletion but instead due to a gain of toxicity induced by an unusual cuprizone:copper complex [44]. Whatever the precise underlying mechanism of the cuprizone-induced toxicity is, it triggers a highly reproducible demyelination of distinct white and gray matter regions. Of note, although demyelination is widespread, it occurs in a region-specific manner. For example, at the level of the rostral hippocampus, the medial part of the corpus callosum is almost entirely demyelinated. In contrast, neighboring white matter tracts such as the cingulum, the fornix, and the hippocampal fimbria are less severely affected. In addition to the white matter tracts, various gray matter areas are affected as well, such as the hippocampus [49], thalamus [50] or neocortex [51].

In the cuprizone model, demyelination is complete after an intoxication period of around 5 weeks (i.e., acute demyelination). In case the animals are provided standard chow, spontaneous, endogenous remyelination occurs, which is complete in a matter of weeks [52]. In contrast, prolonged cuprizone intoxication for 12–13 weeks induces chronic lesions which show a limited endogenous remyelination capacity [42]. Early remyelination is often monitored after a remyelination period of 2 weeks [52]. While the expression of different astrocyte marker proteins, among which are glial fibrillary acidic protein (GFAP), aldehyde dehydrogenase 1 family member L1 (ALDH1L1) or Vimentin, is highly increased in the demyelinated areas, expression levels decrease again after the cessation of cuprizone intoxication. Nevertheless, expression values remain elevated compared to those of control mice, indicating ongoing astrocyte activation during remyelination [53]. Consequently, some studies demonstrated that the modulation of astrocytes impacts myelin repair [54].

Conceptually, there are three sources of remyelinating cells in the cuprizone model. Firstly, the neural stem cells (NSCs) [35,55,56], which reside in the subventricular zone (SVZ) but can migrate into the corpus callosum, striatum, and fimbria. There, these cells can differentiate into NG2-positive non-myelinating and mature myelinating oligodendrocytes. Notably, the number of NSC-derived oligodendrocytes in vivo increased fourfold after a demyelinating lesion in the corpus callosum. This indicates that SVZ cells participate in myelin repair in the adult brain [55]. Of note, in the cuprizone model, SVZ-NSCs are recruited to the white matter tract corpus callosum during the remyelination phase and are capable of forming new oligodendrocytes. When these SVZ-derived NSCs were ablated, animals displayed reduced oligodendrocyte numbers within the lesioned corpus callosum [57]. The next cell type which can give rise to new myelinating oligodendrocytes are the OPCs, also known as NG2 glia. These OPCs are distributed ubiquitously throughout the CNS white and gray matter. In response to a demyelinating insult, OPCs proliferate rapidly and differentiate into re-myelinating oligodendrocytes, contributing to myelin repair [35,58]. Furthermore, the results of some studies suggest that adult oligodendrocytes, which survived the demyelinating insult, can participate in myelin repair [38,59]. No reports are available so far suggesting that Schwann cells participate in the remyelination process in the cuprizone model as well.

Various protocols have been applied to study the effectiveness of novel compounds to promote remyelination in the cuprizone model. After a 5-week cuprizone intoxication period, the corpus callosum is highly populated with OPCs. In case compound treatment is initiated at the time-point when animals are switched back to normal chow, one can easily study its effects on OPC differentiation and remyelination. However, potential OPC-generating effects might be missed if such a protocol is applied. Alternatively, one might initiate the compound treatment at the beginning of week 4, when OPC activation and proliferation starts (compare Figure 1). One should also be aware that after acute cuprizone-induced demyelination, endogenous remyelination occurs spontaneously. This does not allow the study of pro-myelinating effects in the non-supportive environment. Alternatively, one might either use aged animals [60,61] or apply prolonged, chronic cuprizone intoxication [62], where remyelination is significantly delayed.

## 4. Remyelination—A Clinical Perspective

In MS patients, independent of clinically detectable relapses or inflammation visualized by different imaging modalities, there is a continuous accumulation of clinical disability, a phenomenon which was recently termed by Bruce Cree and colleagues [63] “silent progression”. Since the observed disability progression was associated with accelerated brain atrophy, it is likely directly related to the accumulation of irreversible axonal and neuronal damage. There are several studies suggesting that the promotion of remyelination is one option to prevent this neurodegeneration. In the EAE model, an elegant study has shown that accelerated remyelination prevents axonal loss and improves functional recovery [33]. In post mortem MS tissues, it has been demonstrated that the density of acutely damaged axons is high during active and chronic demyelination but low in remyelinated shadow plaques. These observed patterns of axonal pathology in chronic active EAE were qualitatively and quantitatively similar to those found in MS tissues [64]. Furthermore, it has recently been demonstrated that failed remyelination of the nonhuman primate optic nerve leads to axon degeneration, retinal damage, and visual dysfunction [65]. Finally, the results of a recently published longitudinal combined positron emission tomography (PET)/magnetic resonance imaging (MRI) suggest that intralesional remyelination is associated with the microstructural preservation of surrounding tissues [66].

In principle, two different strategies can be followed to promote remyelination. Firstly, by transplanting cell populations promoting myelin repair, and secondly, by interfering with endogenous pathways orchestrating endogenous remyelination (or its failure). Concerning the first strategy, there is only one completed study so far, the “Neural Stem Cell Transplantation in Multiple Sclerosis Patients (STEMS)”, which has been coordinated by Gianvito Martino and colleagues (ClinicalTrials.gov Identifier: NCT03269071). In this phase 1 trial, human fetal-derived neural stem cells were administered intrathecally into progressive MS patients. While the safety of the approach appears to be good, no results have been presented so far concerning the efficacy of this treatment strategy.

A number of different studies are currently being conducted to investigate the potency of drugs to interfere with endogenous remyelination regulators. The results are, so far, heterogeneous. The randomized, double-blind, placebo-controlled, parallel group, phase 2a trial “CCMR One” that investigated the potency of the retinoid X receptor agonist bexarotene failed, mainly due to safety reasons [67]. All bexarotene-treated participants had at least one adverse event. The SYNERGY trial investigated the potency of opicinumab, a fully humanized anti-LINGO-1 antibody, to promote myelin repair. LINGO-1 expression is upregulated in MS lesions, and blockade using antagonistic antibodies or genetic deletion results in increased axonal myelination, both in vitro and in vivo, with amelioration of the disease in the EAE model [68]. Disappointingly, SYNERGY did not meet the primary endpoint, which was the percentage of participants achieving confirmed disability improvement over 72 weeks [69]. To our knowledge, the development of anti-LINGO1 for remyelination in MS has been stopped.

In contrast to these negative outcomes, the results of the ReBUILD trial suggest that clemastine fumarate might be a potential drug supporting endogenous remyelination in MS patients [70]. In this study, relapsing MS patients with chronic (>6 months) demyelinating optic neuropathy that were treated with clemastine, an anti-histamine and anticholinergic medication, demonstrated improved visual-evoked potential latencies. Of note, the potency of this drug to promote remyelination was discovered in vitro, using micropillar arrays as a high-throughput screening platform for novel therapeutics [2]. Other trials investigating the effectiveness of clemastine as a myelin repair therapy are currently ongoing (ClinicalTrials.gov Identifiers NCT05359653 (ReVIVE), NCT05338450 (RESTORE), NCT05131828 (CCMR Two); NCT05338450 (RESTORE), others).

## 5. Findings in the Cuprizone Model

Table 1 lists the studies published since 2016 using the cuprizone de-/remyelination model. A number of very interesting observations have been made using this model. As outlined above, different sources of remyelinating oligodendrocytes exist. NSCs are located in the subventricular zone, which is responsible for the lifelong generation of interneuron subtypes and oligodendrocytes. Using a novel in silico screening approach, the FDA-approved anti-inflammatory corticosteroid medrysone was identified as a potential regulator of NSC-driven remyelination [71]. After 9 weeks of cuprizone intoxication, medrysone promoted the recovery of myelin basic protein expression, the repopulation of the corpus callosum with oligodendrocytes, and the reformation of nodes of Ranvier numbers [54]. Subsequent studies showed that inhibition of the GLI family zinc finger 1 (Gli1) [72] is functionally involved in NSC-driven remyelination. Another study investigated pro-myelinating effects of the antipsychotic drugs haloperidol and clozapine [73]. In 1972, myelin and oligodendrocyte abnormalities had already been described on the ultrastructural level in the brains of schizophrenic patients [74], and these findings have been confirmed and expanded by others [75,76,77]. The preservation of oligodendrocytes and/or the restoration of damaged myelin sheaths might, thus, be an attractive therapeutic concept in schizophrenia. Patergnani and colleagues demonstrated, using the cuprizone model, that both antipsychotic drugs reversed cuprizone-induced MBP loss and myelin fragmentation in the cerebellum, paralleled by improved motor-performance and amelioration of catalepsy signs. Not surprisingly, clinical trials addressing this topic are ongoing.

In the following subchapters we will give a brief insight into recent findings. While it is out of the scope of this review article to discuss all studies conducted so far, we will briefly comment on some compounds and/or strategies identified as potential pro-myelinating agents to be tested in clinical trials. This selection is based on our own expertise working with this model in the past, and on compounds that are currently being evaluated by other groups as potential pro-myelinating agents.

### 5.1. Anti-LINGO-1 Therapy

A major obstacle for successful axon regeneration in the adult CNS arises from inhibitory molecules in CNS myelin debris, which signal through a common receptor complex of neurons consisting of the ligand-binding Nogo-66 receptor (NgR) and two transmembrane co-receptors, p75 and LINGO-1 (Leucine-rich repeat and Immunoglobin-like domain-containing protein 1) [214]. Activation of this receptor complex with, for example, oligodendrocyte myelin glycoprotein, myelin associated glycoprotein (MAG), or neurite outgrowth inhibitor (Nogo) inhibits axonal regeneration.

In 2005, Mi and colleagues reported that LINGO-1 is also expressed by oligodendrocytes [215]. Reduction in LINGO-1 expression via RNAi lentivirus infection or antagonism via the induced expression of a dominant-negative LINGO-1 variant both promoted the differentiation of cultured oligodendrocytes. In contrast, overexpression of full-length LINGO-1 had the opposite effect and inhibited oligodendrocyte differentiation, indicating that endogenous LINGO-1 expression may inhibit oligodendrocyte differentiation and remyelination. Functional experiments revealed that LINGO-1 antagonism promotes oligodendrocyte differentiation via the upregulation of FYN and, consequently, the downregulation of RhoA-GTP. Developmental myelination was accelerated in LINGO-1 knockout mice, compared to wildtypes. Notably, it has been shown that both, oligodendrocyte-derived and axonal-derived LINGO-1, suppresses oligodendrocyte differentiation [216]. Pro-myelinating activities of LINGO-1 antagonism have subsequently been demonstrated in MOG-induced EAE [68], brain slice cultures, LPC-induced focal demyelination and the cuprizone model [106,215]. After some promising results in the RENEW [217] and SYNERGY trial [69], Biogen announced that it is discontinuing the clinical development of opicinumab, an anti-LINGO1 antibody, based on data from the Phase 2 AFFINITY clinical trial.

### 5.2. Clemastine

Clemastine was identified in 2014 using so-called micropillar arrays as a high-throughput screening platform for potential remyelinating therapies [2]. This approach*’*s principle is quantifying OPC-derived versus mature oligodendrocyte-derived membranes wrapping around micropillar arrays of compressed silica. After performing a screen of 1000 bioactive molecules, the authors found clusters of compounds that promoted either the proliferation or differentiation of cultured OPCS, but not both. Among others, the authors identified eight FDA-approved antimuscarinic compounds that significantly enhanced oligodendrocyte differentiation and membrane wrapping, including clemastine. Clemastine is a widely available first-generation anti-histamine with a favorable safety profile. It is used primarily for the symptomatic treatment of allergies and also exhibits antimuscarinic properties. In subsequently performed in vivo experiments using the LPC-model, clemastine treatment enhanced the differentiation of oligodendrocytes and accelerated remyelination. Soon after that initial publication, pro-remyelinating properties of clemastine were also demonstrated in the cuprizone model [218]. In this study, the potency of clemastine was tested to accelerate remyelination after a 6-week cuprizone intoxication period. Clemastine augmented myelin recovery in the corpus callosum, cortex, and hippocampus, as determined by anti-MBP staining intensities, paralleled by higher numbers of CC1+ mature oligodendrocytes. As mentioned above, the results of the ReBUILD trial suggest that clemastine fumarate might be a potential drug supporting endogenous remyelination in MS patients [70]. Of note, results of additional preclinical trials suggest that clemastine might also be beneficial in other neurological disorders, including hypoxic brain injury [219], age-related memory deficits [220], or Alzheimer*’*s disease [221]. Currently, a phase 3 trial (RESTORE; Clemastine Fumarate as Remyelinating Treatment in Internuclear Ophthalmoparesis and multiple sclerosis; NCT05338450) is being conducted to assess the (long-term) efficacy of clemastine fumarate in a clinical model for MS (i.e., in patients with internuclear ophthalmoparesis and MS). The authors expect the final results of their studies in May 2024.

### 5.3. GPR17-Receptor Modulators

In a sentinel study, Maria Abbracchio’s lab from the University of Milan in Italy showed that the expression of GPR17, a receptor for uracil nucleotides and cysLTs (e.g., UDP-glucose and LTD(4)), is expressed in neurons and parenchymal OPCs, and that, upon induction of brain injury using an established focal ischemia model, the expression of GPR17 increases in neurons as well as proliferating OPCs. From a functional point of view, the in vitro exposure of isolated OPCs to the GPR17 endogenous ligands UDP-glucose and LTD(4) promoted the expression of myelin basic protein, suggesting pro-myelinating effects [222]. The same group demonstrated later that the in vivo knockdown of GPR17 by an antisense oligonucleotide strategy during experimental spinal cord injury induction ameliorated disease severity [223]. Finding that Gpr17 overexpression inhibited oligodendrocyte differentiation and maturation both in vivo and in vitro [224] paved the way for considering GPR17 antagonism as a potential remyelinating strategy in MS. Several years later, it was subsequently demonstrated that loss of GPR17, either globally or specifically in oligodendrocytes, led to an earlier onset of remyelination after LPC-induced myelin injury in mice. Similarly, pharmacological inhibition of GPR17 with pranlukast promoted remyelination [225,226]. In the cuprizone model, high GPR17 expression positively correlated with the intrinsic remyelination capacity [227]. The fact that neurons also express GPR17 and GPR17 antagonism ameliorates neuronal damage in different models [228,229] lets us speculate that GPR17 antagonism not just promotes remyelination but, at the same time, might exert neuroprotective properties.

### 5.4. Sphingosine-1-Phosphate Modulators

The sphingosine-1-phosphate receptors (S1PR) system, which consists of five receptor subtypes (from S1PR1 to 5), is involved in various functions, including cell migration, proliferation, and differentiation. S1PR1 receptors are found on the outside of lymphocytes, and their activation triggers lymphocytes to leave lymph nodes and enter the bloodstream, ultimately making their way into the target tissue. Consequently, antagonism of S1PR1 activity blocks the egress of lymphocytes from the lymph nodes and, in consequence, exerts anti-inflammatory activities [230]. In addition, S1PR1 and 5 are expressed by cells of the CNS, as demonstrated by several groups [231,232,233,234]. On a cellular level, S1PR1 is predominately expressed by astrocytes and microglia, whereas cells of the oligodendrocytic lineage are the major cell type expressing S1PR5 in the CNS. Among the available S1P receptor modulators, fingolimod (non-selective, S1PR1-3-4-5), siponimod, ponesimod, and ozanimod have already been approved by regulatory authorities for the treatment of MS [235].

Some reports claim that the modulation of S1PR activities might promote remyelination. Fingolimod, a non-selective S1PR modulator, is a prodrug that has to be activated by phosphorylation via the Sphingosine kinase 2. Consequently, Sphingosine kinase 2 is required for the modulation of lymphocyte traffic by fingolimod [236]. Following cuprizone withdrawal, spontaneous remyelination occurred in wildtype but not in Sphingosine kinase 2^-/-^ mice, and myelin thickness in these mice was found to be reduced with aging [104]. These results suggest that the S1PR-signalling cascade is involved in regulating myelin repair. In line with this assumption, fingolimod showed protective effects in the LPC model [237,238] and EAE [239] but failed to enhance remyelination in the cuprizone model [240,241,242]. No studies have been published to date regarding the pro-remyelinating potency of ozanimod. However, the results of two studies suggest that ozanimod might protect cells of the oligodendrocyte lineage [243,244].

In 2020, the selective S1PR1 and 5 modulator siponimod (trade name Mayzent*^®^*) received EU approval for treating adults suffering from SPMS with disease activity demonstrated by clinical relapses or imaging of inflammatory activity. As treatment with siponimod has an overall stabilizing effect regarding clinical and radiological outcome measures [245], it is discussed whether some of these protective effects might be modulated by the induction of remyelination. Using a Xenopus tadpole screening approach, Mannioui and colleagues identified siponimod among the most efficient molecules favoring remyelination [246]. Furthermore, increased remyelination, determined by evaluations of magnetization transfer ratio and T2-weighted MRT imaging, was also observed in the cuprizone model [97]. In EAE, siponimod prevented the degeneration of synapses after intracerebroventricular infusion [247]. In slice cultures, where the CNS and the peripheral immune system are virtually uncoupled, siponimod attenuated lysophosphatidic choline-mediated demyelination [248], whereas in vivo siponimod increased myelin basic protein levels after lysophosphatidic choline-induced focal demyelination [249]. Finally, our group recently showed that siponimod protects mature oligodendrocytes in an S1PR5-dependent manner [250]. Further studies addressing the regenerative properties of siponimod are currently ongoing in our laboratory, and the results will hopefully be published soon.

### 5.5. Sex Hormones

The observation that males are less likely to develop MS and often have a more severe disease course than females, and the phenomenon that the relapse rate in female MS patients significantly decreases during pregnancy [251], lead to several projects investigating the potency of sex steroids, particularly estrogens, progesterone and testosterone, to ameliorate the MS disease course. In 2013, Hussain and colleagues demonstrated that in castrated male and female mice, testosterone promoted remyelination after chronic cuprizone intoxication. Testosterone also promoted remyelination of cerebellar slice cultures after LPC-induced demyelination [252,253]. Functional experiments further showed that this protective effect of testosterone involves androgen receptor signaling. In 2019, TestOsterone TreatmEnt on Neuroprotection and Myelin Repair in Relapsing Remitting Multiple Sclerosis (TOTEM-RRMS), a phase-2 randomized, placebo-controlled trial, was initiated to investigate potential protective effects of testosterone in RRMS patients (ClinicalTrials.gov Identifiers: NCT03910738). The authors expect the final results of their studies in May 2023.

## 6. Summary and Conclusions

Several pre-clinical and some clinical studies are currently being carried out to test the pro-remyelinating properties of novel compounds. Although stakeholders, investors and regulatory agencies frequently require that compound effectiveness be demonstrated in the EAE model of MS, direct remyelinating properties can be studied straightforwardly using the cuprizone model. There is good reason st believe that novel compounds will be approved in the next decade to prevent disease progression in MS patients.

To date, magnetic resonance imaging for the treatment management of MS primarily relies on T1-weighted, T2-weighted, fluid-attenuated inversion recovery, and gadolinium-based contrast agent-enhanced sequences, which generate tissue and lesion contrast through differences in water content, proton relaxation, and BBB integrity related to inflammation, demyelination, and neurodegeneration. Notably, the visualization of remyelination is much more complex but advances in this field have been made. For example, diffusion-based modalities such as DTI (diffusion tensor imaging) or proton relaxation-based modalities, such as MTI (magnetization transfer imaging), have become widespread, and are arguably the most commonly used proxies for myelin content in clinical trial settings today [254]. An excellent review article addressing this topic has recently been published [255], highlighting the urgent need to further develop and validate these methods to visualize myelin degeneration and repair.

It will be interesting to see whether or not molecules identified as pro-myelinating agents in the cuprizone model demonstrate beneficial effects in future clinical trials. Several factors might play a role in this context. Although lymphocytes are recruited into the demyelinated areas in the cuprizone model [256], T-cell densities are higher in RRMS patients compared to those in the cuprizone model, which might be relevant due to the suppressive impact of Th17 cells on remyelination pathways [257]. Furthermore, B-cells and plasma cells play important roles in MS but not in the cuprizone model. Another important aspect is the time window to be defined to treat MS patients with pro-myelinating compounds. While, in the cuprizone model, the time window to initiate the treatment is well-defined (id est, during or after acute/chronic demyelination), several different lesions with different remyelination kinetics co-exist in MS patients, which makes the situation much more complex. Finally, species-specific differences in oligodendrocyte functions should be considered, as discussed recently in [258]. In fact, in comparison to myelination, non-myelinating functions of oligodendrocytes, such as metabolic support to axons, regulation of axonal and dendritic growth, the regulation of inflammation and angiogenesis, the synthesis of extracellular matrix to form perineuronal nets, or the regulatory impact on blood-brain barrier function are less well-characterized but should be considered in this context.

## Figures and Tables

**Figure 1 ijms-23-16093-f001:**
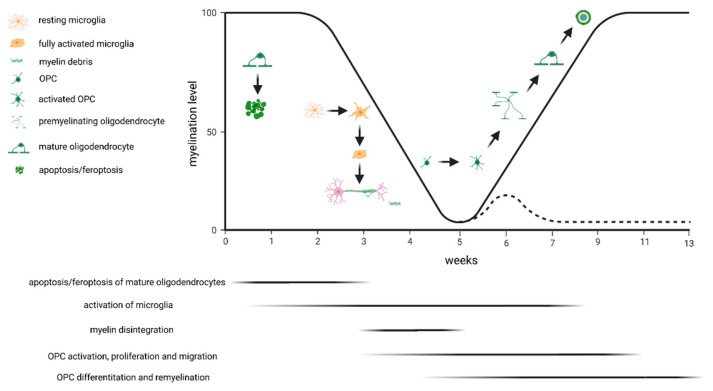
Schematic presentation of the cuprizone-induced pathological changes. The black line indicates the percentage of myelinated fibers in the corpus callosum above the fornix during acute exposure to cuprizone. The dashed line indicates exposure to cuprizone for 13 weeks (i.e., chronic demyelination). The major event at week 1 is apoptosis/feroptosis of the mature oligodendrocytes. In a second step, this triggers the activation of microglia and astrocytes, both phagocytosing the degenerating myelin sheaths. Oligodendrocyte progenitor cells (OPCs) become activated and start to proliferate. If animals are provided normal chow after acute cuprizone-induced demyelination, endogenous remyelination occurs (solid black line) by OPC differentiation and remyelination. If, however, the cuprizone intoxication is continued, remyelination fails (dashed line).

**Table 1 ijms-23-16093-t001:** Summary of publications using the cuprizone model to study which factors regulate remyelination. The terms “cuprizone” and “remyelination” were applied using the PubMed database. Manuscripts from 2016–2022 were included. HC = Histochemistry; IHC = Immunohistochemistry; IF = Immunofluorescence; RT-PCR = reverse transcription polymerase chain reaction; WB = western blotting; TEM = transmission electron microscopy; MRI = magnet resonance imaging; FACS = fluorescence aided cell sorting, (sc)RNA-seq = (single cell)RNA-sequencing, RNA-isH = RNA in situ hybridization. The “*” in column “Citation” indicates that a clinical trial/clinical trials are either ongoing or have already been completed (information retrieved from ClinicalTrials.gov in December 2022; search terms used were “multiple sclerosis” and “the respective compounds”; no pathways were included in this analysis).

Citation	Compound/Intervention	Effects	Region/Method
2022			
[78]	Nox4 (NADPH oxidase 4) deletion	enhanced remyelination ameliorated motor behavioural deficits	Corpus callosum HC, IHC/IF, RT-PCR
[79]	TRPV1 (Transient Receptor Potential Vanilloid 1) receptor activation by capsaicin	enhanced remyelination ameliorated motor behavioural deficits	Corpus callosum HC, IHC/IF, RT-PCR, WB, TEM
[80]	Combination of mesenchymal stem cell transplantation and astrocyte ablation	enhanced remyelination ∙	Corpus callosum HC, IHC/IF, RT-PCR TEM
[81]	Stemazole	enhanced remyelination ameliorated motor behavioural deficits	Corpus callosum HC, IHC/IF
[82]	perilipin-2-deletion	enhanced remyelination	Corpus callosum HC, IHC/IF, RT-PCR
[83]	Huntington disease model R6/2 mice	impaired remyelination	Corpus callosum TEM, mass spectrometry
[84]	Ginsenoside Rg1	enhanced remyelination increased OL numbers	Whole-Brain WB Corpus callosum HC, IHC/IF, TEM
[85]	Ursolic Acid	enhanced remyelination	Brain RT-PCR Corpus callosum HC, IHC/IF
[86]	astragalosides	enhanced remyelination increased numbers of OL	Corpus callosum IHC
[54]	medrysone	enhanced remyelination	Corpus callosum IHC/IF
[87] *	mesenchymal stem cells transplantation and anodal transcranial direct current stimulation	enhanced remyelination increased numbers of OL ameliorated motor behavioural deficits	Corpus callosum HC, IHC/IF, RT-PCR, TUNEL
[88] *	acetyl-L-carnitine	enhanced remyelination ameliorated motor behavioural deficits	Corpus callosum IHC
[89] *	digoxin	enhanced remyelination	Corpus callosum TEM
[90]	lanthionine ketenamine ethyl ester	enhanced remyelination increased numbers of OL	Corpus callosum IHC/IF, RT-PCR, TEM Cortex IHC/IF
[91]	GPR149 deficiency	enhanced remyelination	Corpus Callosum HC, IHC/IF, TEM
[92]	Protein Arginine Methyltransferase 1 deletion	impaired remyelination decreased numbers of OPC	Brain scRNA-seq Corpus callosum HC, IHC/IF, TEM
[93]	tenascin-C or tenascin-R deletion	enhanced remyelination	Corpus callosum HC, IHC/IF, RT-PCR, TEM
[94]	aquatic exercise OR methylprednisolone	enhanced remyelination	Corpus callosum HC, IHC/IF Hippocampus RT-PCR
[95]	acer truncatum oil	enhanced remyelination ameliorated behavioural deficits increased numbers of OL	Corpus Callosum HC, IHC/IF, TEM
[96]	Liraglutide	enhanced remyelination	Whole Brain Weigth, WB Corpus callosum HC, IHC/IF
[97] *	Siponimod	enhanced remyelination T2-weighted signal intensity decrease	Corpus callosum HC, IHC/IF, MRI
[98]	Baicalin	enhanced remyelination ameliorated behavioural deficits	Corpus callosum HC, IHC/IF, RT-PCR, TEM
[99] *	(R)-ketamine	enhanced remyelination	Corpus callosum IHC/IF
[100] *	Mesenchymal stem cell transplantation	enhanced remyelination increased numbers of OL	Corpus callosum HC, IHC/IF, RT-PCR, TEM
[101]	Ethoxymethyl ether salvinorin b	enhanced remyelination increased numbers of OL	Corpus callosum IHC/IF, TEM
[20]	Injection of citrullinated myelin	impaired remyelination	Cortex IHC/IF
[102]	Loading modified exosomes (expressing the ligand of platelet-derived growth factor receptor *α* (pdgfr*α*)) with Bryostatin-1	enhanced remyelination	Whole-Brain RT-PCR Corpus callosum HC, IHC/IF, TEM
2021			
[103]	CDP-choline	enhanced remyelination increased numbers of OL	Corpus callosum HC, IHC/IF
[104]	Sphingosine kinase 2 deletion	impaired remyelination	Corpus callosum HC, IHC/IF, WB Cortex IHC/IF, WB
[105]	Bacillus coagulans treatment	enhanced remyelination ameliorated behavioural deficits	Whole-Brain RT-PCR Corpus callosum HC, IHC/IF
[106] *	LINGO-1 antibody	enhanced remyelination ameliorated behavioural deficits	Corpus callosum IHC/IF
[107]	Poly (ADP-ribosyl) polymerase 1 (PARP1) depletion	impaired remyelination lower numbers of OL	Corpus callosum HC, IHC/IF,
[108] *	Ozanimod	no effect on remyelination	Corpus callosum, Cortex, Hippocampus IHC/IF, MRI
[109] *	Hydroxychloroquine	increased numbers of OPC enhanced remyelination	Corpus callosum, SVZ HC, IHC/IF
[110]	Blockade of Bone Morphogenetic Protein-2/4	enhanced remyelination higher numbers of OL	Corpus callosum IHC/IF
[73]	Antipsychotic drugs haloperidol and clozapine	enhanced remyelination ameliorated motor behavioural deficits	Whole-Brain WB Cerebellum IHC/IF
[111] *	N-acetylcysteine	enhanced remyelination	Corpus callosum HC, IHC/IF, TEM
[112]	Apamin	increased numbers of OL	Corpus callosum IHC/IF
[113]	genetic or pharmacological ISR (integrated stress response) enhancement	enhanced remyelination increased numbers of OL	Corpus callosum IHC/IF, TEM
[114]	danazol, parbendazole	enhanced remyelination	Corpus callosum IHC/IF
[115]	Catalpol	enhanced remyelination ameliorated behavioural deficits increased numbers of OL	Corpus callosum HC, IHC/IF, TEM Whole-Brain RT-PCR, WB
[116]	Mertk-kockout	impaired clearance of myelin debris impaired remyelination	Corpus callosum HC, IHC/IF, RT-PCR, TEM, FACS, scRNA-seq
[117]	Act-1004-1239	enhanced remyelination increased numbers of OL	Corpus callosum HC, IHC/IF,
[118]	nalfurafine	enhanced remyelination	Corpus callosum TEM
[119]	N-acetylglucosamine	enhanced remyelination ameliorated motor behavioural deficits	Corpus callosum IHC/IF, TEM
2020			
[120]	QKI (QUAKING)-knockout	impaired remyelination	Corpus callosum HC, IHC/IF
[121]	Nestorone	increased numbers of OL	Corpus callosum RT-PCR, WB Hippocampus RT-PCR, WB
[122] *	Coenzyme Q10	enhanced remyelination higher numbers of OL	Corpus callosum HC, RT-PCR, WB
[123] *	Guanabenz	no impact on remyelination or OL numbers	Corpus callosum HC, IHC/IF
[124]	forkhead box G1 (Foxg1) deletion	enhanced remyelination higher numbers of OL	Corpus callosum HC, IHC/IF
[125]	Tropomyosin receptor kinase B (Trkb) receptor deletion	impaired remyelination	Corpus callosum HC, IHC/IF, TEM, WB
[126] *	Elvitegravir and raltegravir	impaired remyelination	Corpus callosum HC, IHC/IF
[127]	ferritin heavy subunit (Fth) deletion	impaired remyelination lower numbers of OL	Whole-Brain WB Corpus callosum, Cortex, Striatum IHC/IF Corpus callosum TEM
[128] *	Hydroxychloroquine	enhanced remyelination	Whole-Brain RT-PCR Corpus callosum HC
[129]	Anacardic acid	enhanced remyelination	Corpus callosum HC, IHC/IF, TEM
[130]	Triggering receptor expressed on myeloid cells 2 (TREM2) agonistic antibody	accelerated myelin debris removal by microglia enhanced remyelination increased numbers of OL	Corpus callosum HC, IHC/IF, RT-PCR, TEM, FACS Hippocampus RT-PCR
[131]	Cellular Communication Network Factor 3 (CCN3) knockout	no effect on remyelination	Corpus callosum HC, IHC/IF Lateral septum IHC/IF
[132] *	Calorie restriction	enhanced remyelination increased numbers of OL ameliorated motor behavioural deficits	Corpus callosum HC, IHC/IF, RT-PCR, TUNEL
[133] *	Ehp-101	enhanced remyelination	Corpus callosum HC, IHC/IF Cortex HC, IHC/IF
[134] *	Transplantation of induced neural stem cells	normalized imaging abnormalities ameliorated behavioural deficits	Corpus callosum IHC/IF, MRI,RNA-isH Cortex IHC/IF
[135]	TMEM106B deletion	impaired remyelination	Corpus callosum HC, IHC/IF, TEM, WB
[136]	Transplantation of human glial progenitor cells	enhanced remyelination	Corpus callosum IHC/IF, RNAseq
[137]	Learning a forelimb reach task	Motor learning promotes the participation of pre-existing mature oligodendrocytes in remyelination	Cortex IHC/IF, live-imaging
[138]	Injection of Sox10 overexpressing virus into hippocampus	enhanced remyelination ameliorated behavioural deficits	Hippocampus IHC/IF, TEM, WB
[139]	Ursolic acid	enhanced remyelination	Corpus callosum HC, IHC/IF, TEM
[140]	P2x7 receptor blockade	no effect on remyelination	Whole-Brain RT-PCR, WB Corpus callosum HC, IHC/IF
[141]	Alpha Synuclein deficiency	no effect on remyelination	Corpus callosum HC, IHC/IF, MRI
[142]	Cav1.2 channels, nimodipine	enhanced remyelination increased numbers of OL	Corpus callosum, Cortex, Cerebellum IHC/IF, RT-PCR, WB Corpus callosum TEM
[143] *	Gold nanocrystals	enhanced remyelination	Corpus callosum IHC/IF, TEM
[144]	Ginkgolide B	ameliorated behaviour abnormalities enhanced remyelination	Whole-Brain WB Corpus callosum, Striatum, Cortex HC, IHC/IF
[145] *	Cannabinoid (WIN-55,212-2)	impaired remyelination	Corpus callosum HC, IHC/IF, RT-PCR, WB
[146]	Protease Activated Receptor 1 (PAR1) deletion	enhanced remyelination higher numbers of OL ameliorated motor behavioural deficits	Corpus callosum IHC/IF
[147]	Glycyrrhizic acid	enhanced remyelination	Corpus callosum HC, IHC/IF
2019			
[148]	Apamin	enhanced remyelination	Corpus callosum HC
[149] *	Fingolimod	no effect on white matter remyelination increased numbers of OL in the cortex	Whole-Brain Proteomics Corpus callosum HC, IHC/IF
[150] *	Melatonin	enhanced remyelination (specifically in male mice)	Brainstem WB
[151] *	Laquinimod	enhanced remyelination increased numbers of OL	Corpus callosum HC, IHC/IF
[152]	CXCR2 antagonism (via compound 2)	enhanced remyelination ameliorated behaviour abnormalities	Corpus callosum HC, IHC/IF, RT-PCR, WB Hippocampus WB
[153]	Phloroglucinol derivative compound 21	enhanced remyelination ameliorated behaviour abnormalities	Corpus callosum HC, IHC/IF, WB
[154] *	Metformin	increased numbers of OL	Corpus callosum IHC/IF, RT-PCR, WB
[155]	Disruption of Sema3A/Plexin-A1 inhibitory signalling in oligodendrocytes	enhanced remyelination ameliorated behaviour abnormalities	Corpus callosum HC, IHC/IF, MRI
[156]	Trkb Agonist LM22A-4	enhanced remyelination increased numbers of OL	Corpus callosum IHC/IF, TEM
[157]	low dose TLR2 ligands to induce systemic TLR2 tolerance	enhanced remyelination	Corpus callosum IHC/IF, TEM
[158]	Clozapine	enhanced remyelination ameliorated behaviour abnormalities	Corpus callosum HC, IHC/IF
[159] *	Erβ-ligand	enhanced remyelination	Corpus callosum IHC/IF, TEM, RNAseq
[160]	Inhibiting Bone Morphogenetic Protein 4 Type I Receptor by LDN-193189	enhanced remyelination increased numbers of OL	Corpus callosum IHC/IF, TEM, Live-imaging
[161] *	Donepezil	enhanced remyelination increased numbers of OL	Corpus callosum HC, IHC/IF, TEM
[162]	PD0325901 (MEK (MAPK kinase) inhibitor)	enhanced remyelination	Corpus callosum HC, IHC/IF, TEM
[163]	La-aminoadipate mediated astrocyte depletion	enhanced remyelination ameliorated behaviour abnormalities	Corpus callosum HC, IHC/IF, RT-PCR, TEM
[164]	Methylthioadenosine, delivered as solid lipid nanoparticles	enhanced remyelination ameliorated behaviour abnormalities	Corpus callosum HC
[165] *	D-aspartate	enhanced remyelination ameliorated behaviour abnormalities	Corpus callosum IHC/IF, TEM, WB
[166]	Leonurine	enhanced remyelination	Corpus callosum HC, IHC/IF
2018			
[167]	benztropine	enhanced remyelination	Corpus callosum HC, IHC/IF
[168]	N-Phenylquinazolin-2-amine	enhanced remyelination	Corpus callosum HC, IHC/IF
[169] *	Vitamin K	no effect on remyelination	Whole-Brain Lipidomics Corpus callosum HC, IHC/IF
[170] *	Adenosine	enhanced remyelination ameliorated behaviour abnormalities	Cortex,Hippocampus IHC/IF, WB
[171]	BDNF-mimetic	enhanced remyelination, increased numbers of OL	Corpus callosum IHC/IF, TEM
[172] *	recombinant human-derived monoclonal IgM antibody rHIgM22	enhanced remyelination ameliorated behaviour abnormalities	Hippocampus HC, IHC/IF
[173] *	Dimethyl fumarate	reversed electrophysiological abnormalities ameliorated behaviour abnormalities no effects on remyelination	Brain slices Electrophysiology IHC/IF
[174]	MCHII knockout in microglia	no effect on remyelination	Corpus callosum, Interposedcerebellar nucleus IHC/IF, MRI Corpus callosum FACS, RNAseq
[175]	SAG injection (agonist of canonical and type II non-canonical Hedgehog signaling pathways)	enhanced remyelination	Corpus callosum, Cortex IHC/IF
[176]	Neuroblast reprogramming; i.e., forced expression of transcription factors OLIG2 and SOX10	enhanced remyelination increased numbers of OL	Corpus callosum IHC/IF, TEM
[177]	Microrna-146a knockout	no effect on remyelination	Corpus callosum HC, IHC/IF, RT-PCR, WB, Proteomics
[178]	colony-stimulating factor 1 receptor kinase inhibitor; BLZ945	enhanced remyelination increased numbers of OL	Corpus callosum, Cortex, Striatum HC, IHC/IF, MRI
[179] *	Vitamin C	enhanced remyelination increased numbers of OL	Corpus callosum HC, IHC/IF, TEM
2017			
[180]	histamine receptor-3 inverse agonist; GSK247246	enhanced remyelination	Corpus callosum HC, IHC/IF, TEM Cortex HC
[181]	Protamine	enhanced remyelination	Corpus callosum HC, IHC/IF, RT-PCR, Micro-CT
[182]	Contactin-2 deletion	no effect on remyelination	Corpus callosum HC, IHC/IF, TEM Brain-slices Electrophysiology
[183]	myosin ID deletion	impaired remyelination	Corpus callosum HC, IHC/IF, RT-PCR
[184] *	Green tea epigallocatechin-3-gallate	increased PLP and Olig1 mRNA expression	Cortex RT-PCR
[185]	Gas6^-/-^ Axl^-/-^ double knockout	impaired remyelination	Corpus callosum HC, IHC/IF, RT-PCR, TEM
[186]	Ganglioside Gd1a	overcomes inhibition of (re)myelination by aggregated fibronectin	Corpus callosum IHC/IF, RNA-isH
[187]	L-type voltage-gated calcium channel Cav1.2 deletion	impaired remyelination impaired OPC differentiation lower numbers of OL	Whole-Brain,Corpus callosum WB Corpus callosum, Cortex, Striatum HC, IHC/IF Corpus callosum TEM
[188]	NF-κb inactivation in oligodendrocytes	impaired remyelination	Corpus callosum IHC/IF, TEM
[189]	Ncam1 or St8sia2 deletion	impaired remyelination reduced numbers of OL more severe behaviour abnormalities	Corpus callosum HC, IHC/IF
[190]	2-carba-cyclic phosphatidic acid	enhanced remyelination ameliorated behaviour abnormalities	Corpus callosum HC, IHC/IF, RT-PCR, TEM
[191]	Acid sphingomyelinase deficiency	enhanced remyelination increased numbers of OL	Corpus callosum HC, IHC/IF, RT-PCR, WB
[192] *	recombinant human-derived monoclonal IgM antibody rHIgM22	enhanced remyelination increased numbers of OL	Corpus callosum HC, IHC/IF
[193]	Leukemia/lymphoma-related factor (LRF) deletion	impaired remyelination	Corpus callosum IHC/IF, RNA-isH
[194]	XPro1595, a selective inhibitor of soluble TNF	enhanced remyelination enhanced phagocytosis ameliorated behaviour abnormalities	Whole-Brain RT-PCR Corpus callosum HC, IHC/IF, TEM
[195]	TnP	enhanced remyelination	Corpus callosum HC, IHC/IF
[196]	Transplantation of mir-219-overexpressing oligodendrocyte precursor cells	enhanced remyelination ameliorated behaviour abnormalities	Corpus callosum HC, IHC/IF, TEM
[197]	Stereotactic injections	no impact on remyelination	Corpus callosum, Cortex IHC/IF
[198]	Electroacupuncture	enhanced remyelination ameliorated behaviour abnormalities	Corpus callosum HC, IHC/IF, RT-PCR, RNA-seq, WB
[199] *	Dietary cholesterol	enhanced remyelination ameliorated behaviour abnormalities increased numbers of OL	Corpus callosum HC, IHC/IF, RT-PCR, TEM
[200] *	Feeding fat-1 mice with omega-3 fatty acids	no effect on remyelination	Whole-Brain Chromatography Corpus callosum HC
[201]	1,4-dideoxy-1,4-imino-d-arabinitol (DAB)	impaired remyelination ∙	Corpus callosum HC, IHC/IF
[202]	Gal-3 (Lgals3^-/-^)-deletion	impaired remyelination	Corpus callosum HC, IHC/IF, TEM, WB
2016			
[203]	Cord blood monocyte-derived cell therapy product (DUOC-01) in NOD/SCID/IL2Rγ^null^ mice	enhanced remyelination enhanced OPC proliferation	Corpus callosum HC, IHC/IF, TEM
[204]	Social isolation	impaired remyelination	Cortex IHC/IF, RT-PCR, TEM
[205]	Cyclin-dependent kinase inhibitor flavopiridol	enhanced remyelination ameliorated behaviour abnormalities	Cortex RT-PCR Cortex, SVZ IHC/IF
[206] *	3-day cycles of a fasting mimicking diet (FMD)	enhanced remyelination increased numbers of OL	Corpus callosum HC, IHC/IF
[207] *	Intraventricular injections of mesenchymal stem cells	enhanced remyelination reversed electrophysiological abnormalities	Corpus callosum IHC/IF, TEM, MRI Brain slices Electrophysiology
[208]	CNS Penetrant CXCR2 Antagonist	enhanced remyelination	Corpus callosum HC
[209]	Resveratrol	enhanced remyelination ameliorated behaviour abnormalities	Whole-Brain HC, RT-PCR, Enzyme activity
[210] *	Triiodothyronine	enhanced OPC differentiation enhanced remyelination	Whole-Brain RT-PCR Corpus callosum IHC/IF, TEM
[211]	Thymosin beta4	enhanced remyelination enhanced OPC proliferation	Corpus callosum IHC/IF, WB
[212] *	Quetiapine	enhanced remyelination	Corpus callosum HC, IHC/IF
[213] *	Triiodothyronine	enhanced remyelination increased numbers of OL	Corpus callosum IHC/IF, RT-PCR, WB

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
