# Peer review of "Remyelination in Multiple Sclerosis: Findings in the Cuprizone Model"

_ijms, 2022, doi:10.3390/ijms232416093_

Round 1
Reviewer 1 Report
The authors report a narrative review on the utility of the cuprizone’s animal model for the development of drugs that promote remyelination.
I appreciate the table which summarizes the contributions generated by the use of this model. Drug development from preclinical to clinical evaluation in humans is briefly discussed in Chapter 5.
I have several remarks to improve the manuscript:
1. The introduction is very general, and perhaps deserves to be more centered on the oligodendrocyte (stage of development…) and the tools for evaluating remyelination in animals and humans, which is a big challenge today.
2. I would recommend adding the publications using the cuprizone model to test the effect of hormones on remyelination. This avenue for promoting remyelination is currently being tested in a clinical trial and comes from the cuprizone model : Hussain et al, PMID: 23365095. ClinicalTrials.gov Identifier: NCT03910738.
3. I would also recommend adding an updated description of clemastine development as a phase 3 is currently underway : ClinicalTrials.gov Identifier: NCT05338450
4. Finally I would be very happy to see in the table for each molecule the level of development of the drug in humans (phase 1 or 2 or 3…) or if the development has been stopped and for what reason (efficacy, tolerance… )
5. A brief comment on why the molecules work in the cuprizone animal model and not in human clinical trials would be welcome for clinicians reading the article.
Author Response
We would like to thank the reviewer for his/her effort to read and comment on our joint manuscript. Your comments are highly appreciated!
Question 1. The introduction is very general, and perhaps deserves to be more centered on the oligodendrocyte (stage of development…) and the tools for evaluating remyelination in animals and humans, which is a big challenge today.
Answer 1: Thank you for this comment. As suggested by the kind reviewer, we have adopted the manuscript. Firstly, very general aspects of the introduction have been deleted. Secondly, we have added some aspects of oligodendrocyte biology (see Section 2) and highlighted the urgent need to further develop and validate imaging modalities to visualize myelin degeneration and repair (see Section 6).
Question 2. I would recommend adding the publications using the cuprizone model to test the effect of hormones on remyelination. This avenue for promoting remyelination is currently being tested in a clinical trial and comes from the cuprizone model : Hussain et al, PMID: 23365095 [Titel anhand dieser Pubmed-ID in Citavi-Projekt übernehmen] . ClinicalTrials.gov Identifier: NCT03910738.
Answer 2: Thank you for this advice. We have included this in Section 5.5
Question 3. I would also recommend adding an updated description of clemastine development as a phase 3 is currently underway : ClinicalTrials.gov Identifier: NCT05338450
Answer 3: Thank you for this advice. We have included this in Section 5.2
Question 4. Finally I would be very happy to see in the table for each molecule the level of development of the drug in humans (phase 1 or 2 or 3…) or if the development has been stopped and for what reason (efficacy, tolerance… )
Answer 4: Thank you for this comment. For most of the compounds listed in table 1, no clinical trials have been completed so far (at least to our knowledge). One should be aware that the table lists results from the last six years, and the time span might be too short to have already results of (ongoing) clinical trials. We hope the reviewer can follow our argumentation.
Question 5. A brief comment on why the molecules work in the cuprizone animal model and not in human clinical trials would be welcome for clinicians reading the article.
Answer 5: Thank you for this fruitful comment. We have adopted the manuscript accordingly.
Reviewer 2 Report
This review gives an overview of pharmacological approaches to stimulate remyelination in the cuprizone model of multiple sclerosis. The tabulation of these approaches in one table is useful and constitutes a useful resource. 4 pharmacological stimulation strategies for remyelination out of 136 listed approaches were chosen for more in depth description.
Concerns:
1) The schematic figure attempting to depict the dynamic hall marks of the demyelination and remyelination dynamics as presented is difficult to decipher and misleading.
ideas for Fig1 improvements:
use arrows/vectors underneath time course time line to depict individual key events (ie. onset and duration of OL death, onset and duration of OPC activation, onset and duration of OPC differentiation into OL, onset and duration of axon myelination, etc....
increase font size and size of symbols in Fig.
2) It is not clear why the four approaches have been chosen for greater discussion. Please describe on what this selection is based (i.e. author expertise and interest, current topics, corroboration by multiple labs, underlying logic of proposed mechanisms, etc.)
3) The table gives one reference per approach. As presented the table might lead to misinterpretations and might be misleading. The utility of the table and the entire review could be vastly improved by defining the criteria for choosing the given reference and by including further details in the table.
Please state why the given reference was chosen (first to report, mentioned in prior review etc.).
Please add a column stating where remyelination was observed/quantified (ie. CC, parenchyma and anatomic locus, spinal cord tracts, periphery).
Please add a column whether finding was validated by an independent study.
typos:
line 68: reference to general textbook seems out of style and place.
line 74: "one single section of an axon": of one axon or several axons? please specify.
line 95: "called named" delete one of the verbs
line 135: "both": there are three listed?
line 140: throughout manuscript replace "chopper" with copper.
line 194: "at the time point when normal chow is provided": meaning "when animals are switched back to normal chow?
line 213: "get" --> "become"
line 284: "approved" --> "confirmed"
line 294: "scope the review" --> scope of the review
line 415: "currently neeing carried out": not clear what's meant.
line 419: "so" --> "to"
Author Response
We would like to thank the reviewer for his/her effort to read and comment on our joint manuscript. Your comments are highly appreciated!
Question 1: The schematic figure attempting to depict the dynamic hallmarks of the demyelination and remyelination dynamics as presented is difficult to decipher and misleading. ideas for Fig1 improvements: use arrows/vectors underneath time course time line to depict individual key events (ie. onset and duration of OL death, onset and duration of OPC activation, onset and duration of OPC differentiation into OL, onset and duration of axon myelination, etc.... increase font size and size of symbols in Fig.
Answer 1: Thank you for this comment. We have substantially revised the figure, taking into account your valuable suggestions.
Question 2: It is not clear why the four approaches have been chosen for greater discussion. Please describe on what this selection is based (i.e. author expertise and interest, current topics, corroboration by multiple labs, underlying logic of proposed mechanisms, etc.)
Answer 2: Thank you for this valuable comment. Following the suggestion of the kind reviewer we have modified section 5 of the revised version of the manuscript and added the following statement: “In the following subchapters we will give a brief insight in recent findings. While it is out of the scope of this review article to discuss all studies conducted so far, we will briefly comment on some compounds and/or strategies identified as potential pro-myelinating agents to be tested in clinical trials. This selection is based on the one hand on our own expertise working with this model in the past, on the other hand on compounds that are currently evaluated by other groups as potential pro-myelinating agents.”
Beyond, based on the comments of reviewer #1, we have added a new subchapter describing efforts to evaluate the potency of sex hormones to exert beneficial effects in MS. In particular we state the following: “ The observations that males are less likely to develop MS and often have a more severe disease course than females, and the phenomenon that the relapse rate in female MS patients significantly decreases during pregnancy [119] lead to several projects inves-tigating the potency of sex steroids, particularly estrogens, progesterone and testos-terone, to ameliorate the MS disease course. In 2013, Hussain and colleagues demon-strated that in castrated male and female mice, testosterone promoted remyelination after chronic cuprizone-intoxication. Testosterone as well promoted remyelination of cerebellar slice cultures after LPC-induced demyelination [120, 121]. Functional ex-periments further showed that this protective effect of testosterone involves androgen receptor signaling. In 2019, TestOsterone TreatmEnt on Neuroprotection and Myelin Repair in Relapsing Remitting Multiple Sclerosis (TOTEM-RRMS), a phase 2 random-ized, placebo-controlled trial, was initiated to investigate potential protective effects of testosterone in RRMS patients (ClinicalTrials.gov Identifiers: NCT03910738). In May 2023 the authors expect the final results of their studies. “
Question 3: The table gives one reference per approach. As presented the table might lead to misinterpretations and might be misleading. The utility of the table and the entire review could be vastly improved by defining the criteria for choosing the given reference and by including further details in the table. Please state why the given reference was chosen (first to report, mentioned in prior review etc.). Please add a column stating where remyelination was observed/quantified (ie. CC, parenchyma and anatomic locus, spinal cord tracts, periphery). Please add a column whether finding was validated by an independent study.
Answer 3: Thank you very much for your kind suggestions. We have substantially revised the table now stating the region of interest (i.e., anatomical locus) as well as which methods were applied to evaluate the extent of myelination. There was no bias in choosing any reference. To identify the relevant literature, an electronic search was performed using PubMed applying the search terms “remyelination” AND “Cuprizone”. Articles which were published after 2015 were included. The search yielded 364 articles, of which those given fulfilled the eligibility criteria (i.e., indeed investigating factors regulating remyelination and not demyelination in the cuprizone model) and were included in this systematic review. We are sorry that this was not pointed out. We have adopted the revised version of the manuscript accordingly.
Question 4: typos; line 68: reference to general textbook seems out of style and place; line 74: "one single section of an axon": of one axon or several axons? please specify; line 95: "called named" delete one of the verbs; line 135: "both": there are three listed?; line 140: throughout manuscript replace "chopper" with copper; line 194: "at the time point when normal chow is provided": meaning "when animals are switched back to normal chow; line 213: "get" --> "become"; line 284: "approved" --> "confirmed"; line 294: "scope the review" --> scope of the review; line 415: "currently neeing carried out": not clear what's meant; line 419: "so" --> "to"
Answer 4: Thank you very much for carefully reading this drafted manuscript. We have adopted the manuscript accordingly.
Round 2
Reviewer 1 Report
Thanks to the authors for their responses.
Unfortunately, they did not respond to the development phase in humans of the molecules listed in table 1. Some of them have been tested in humans (siponimod, ozanimod, laquiniqmod, antiLINGO-1, mesenchymal stem cells ...).
The authors added a paragraph on the difficulty of assessing remyelination in humans compared to the cuprizone model. This is only part of the explanations that highlight the discrepancy in remyelination outcomes in human compared to what observed in the cuprizone animal model. I would recommend completing this argument further (role of inflammation in the cuprizone vs human model, time window to be defined to treat people with MS, type of oligodendrocyte in both species...)
Author Response
Reviewer #2
We would like to thank the reviewer for his/her effort to read and comment on our joint manuscript. Your comments are highly appreciated!
Question 1: Unfortunately, they did not respond to the development phase in humans of the molecules listed in table 1. Some of them have been tested in humans (siponimod, ozanimod, laquiniqmod, antiLINGO-1, mesenchymal stem cells ...).
Answer 1: Thank you for this comment. We currently prepare another review article where we discuss all the compounds that are currently tested in clinical trials or have been tested previously. We hope that we will publish this article early in 2023. Nevertheless, we now highlight by a “*” in the table (first column, together with the provided citation) that clinical trials are either ongoing or completed (based on ClinicalTrials.gov). The table legend was modified accordingly.
Question 2: The authors added a paragraph on the difficulty of assessing remyelination in humans compared to the cuprizone model. This is only part of the explanations that highlight the discrepancy in remyelination outcomes in human compared to what observed in the cuprizone animal model. I would recommend completing this argument further (role of inflammation in the cuprizone vs human model, time window to be defined to treat people with MS, type of oligodendrocyte in both species...)
Answer 2: Thank you very much for this comment. We have modified the section according to your suggestions. We now state the following in the revised version of the manuscript: “It will be interesting to see in the future whether or not molecules identified as pro-myelinating agents in the cuprizone model demonstrate beneficial effects in clinical trials. Several factors might play a role in this context. Although lymphocytes are recruited into the demyelinated areas in the cuprizone model [124], T-cell densities are higher in RRMS patients compared to the cuprizone model, which might be relevant due to the suppressive impact of Th17 cells on remyelination pathways [125]. Furthermore, B-cells and plasma cells play important roles in MS but not in the cuprizone model. Another important aspect is the time window to be defined to treat MS patients with pro-myelinating compounds. While in the cuprizone model the time window to initiate the treatment is well defined (id est, during or after acute/chronic demyelination), in MS patients several different lesions with different remyelination kinetics co-exist, which makes the situation much more complex. Finally, species-specific differences in oligodendrocyte functions should be considered, as discussed recently in [126]. In fact, in comparison to myelination, non-myelinating functions of oligodendrocytes, such as metabolic support to axons, regulation of axonal and dendritic growth, the regulation of inflammation and angiogenesis, the synthesis of extracellular matrix to form perineuronal nets, or the regulatory impact on blood-brain barrier function are less well characterized but should be considered in this context.”